# Soil Bioassay for Detecting *Magnaporthiopsis maydis* Infestation Using a Hyper Susceptible Maize Hybrid

**DOI:** 10.3390/jof6030107

**Published:** 2020-07-13

**Authors:** Ofir Degani, Danielle Regev, Shlomit Dor, Onn Rabinovitz

**Affiliations:** 1Plant Sciences Department, Migal Galilee Research Institute, Tarshish 2, Kiryat Shmona 11016, Israel; linkar45@gmail.com (D.R.); dorshlomit@gmail.com (S.D.); onnrab@gmail.com (O.R.); 2Faculty of Sciences, Tel-Hai College, Upper Galilee, Tel-Hai 12210, Israel

**Keywords:** bioassay, *Cephalosporium maydis*, crop protection, fungus, *Harpophora maydis*, late wilt, *Magnaporthiopsis maydis*, sensitive cultivar, soil assay

## Abstract

*Magnaporthiopsis maydis* is the causal agent of severe maize late wilt disease. Disease outbreak occurs at the maize flowering and fruit development stage, leading to the plugging of the plant’s water vascular system, resulting in dehydration and collapse of the infected host plant. The pathogen is borne by alternative hosts, infected seeds, soil, and plant residues and gradually spreads to new areas and new countries. However, no soil assay is available today that can detect *M. maydis* infestation and study its prevalence. We recently developed a molecular quantitative Real-Time PCR (qPCR) method enabling the detection of the *M. maydis* DNA in plant tissues. Despite the technique’s high sensitivity, the direct examination of soil samples can be inconsistent. To face this challenge, the current work demonstrates the use of a soil bioassay involving the cultivation of a hyper-susceptible maize genotype (Megaton cultivar, Hazera Seeds Ltd., Berurim MP Shikmim, Israel) on inspected soils. The use of Megaton cv. may facilitate pathogen establishment and spread inside the plant’s tissues, and ease the isolation and enrichment of the pathogen from the soil. Indeed, this cultivar suffers from severe dehydration sudden death when grown in an infested field. The qPCR method was able to accurately and consistently identify and quantify the pathogen’s DNA in an in vitro seed assay after seven days, and in growth-chamber potted plants at as early as three weeks. These results now enable the use of this highly susceptible testing plant to validate the presence of the maize late wilt pathogen in infested soils and to evaluate the degree of its prevalence.

## 1. Introduction

The causal agent of the maize late wilt disease is the soil- and seed-borne pathogen *Magnaporthiopsis maydis* (former names—*Harpophora maydis* and *Cephalosporium maydis*) [1]. The pathogen penetrates the plant’s root system at an early stage of the growth period and may cause delayed or reduced seedling sprouting [2,3]. The most prominent disease symptoms appear months later [4,5]. During this time, the pathogen gradually spreads upwards inside the host plant’s water vascular system without any visible symptoms. At the age of 50–60 days (depending on the maize cultivar susceptibility degree and environmental growth conditions, especially the watering regime) [6,7], the plant’s lower parts (mainly the leaves) begin to lose their green color. This begins a relatively rapid dehydration process that peaks 10–20 days later, resulting in the plant’s death [8]. The disease is considered to be the predominant devastating maize disease in Egypt and Israel, and a serious threat to other areas, including India [9], Hungary [10], Romania [11], Spain and Portugal [12], and Nepal [13].

The pathogen can survive in the absence of susceptible maize hosts over extended periods in the soil as sclerotia, spores, or hyphae on plant remains [14]. The sclerotia consist of a packed fungal mycelium surrounded by a thick tissue enriched with high melanin content that protects it from UV light [15]. *M. maydis* can also survive by developing inside an alternative host plant. To date, these include *Lupinus termis* L. (lupine) [16], *Citrullus lanatus* (watermelon), *Gossypium hirsutum L.* (cotton), and *Setaria viridis* (green foxtail) [17,18].

A traditional method for determining soil infestation level includes soil plating onto semi-selective media and counting colonies of the fungus following incubation for several weeks. However, this timetable can be problematic when maize growers have to make planting decisions; therefore, methods that could guarantee more quickly and accurate results would be advantageous to economic production. Another approach to provide a faster technique of fungal pathogen detection and quantification in the soil is the use of polymerase chain reaction (PCR) molecular identification [19]. Customary PCR tests using nested or single amplifications have been suggested for detecting other fungi (summarized, for example, for *Verticillium dahliae* by [20]). A TaqMan probe assay has been developed lately to target *M. maydis* [21] and could be used for the same purpose, as demonstrated for the potato pathogen, *Synchytrium endobioticum* [22].

However, in the case of *M. maydis*, the goal of developing an effective soil assay has yet to be achieved. This assay may be challenging to develop due to low quantities of the fungus in the soil and the scattered nature of the disease (that spreads in patches in the field). To make this task even harder, DNA extracted from soil samples may include PCR inhibitors [23]. With the gradual canceling of soil disinfecting using methyl bromide [24], lengthier intervals are needed between soil treatments, using alternative fungicides, and sowing. This challenge, together with the increase in production of organic cultivars, necessitates an urgent need for an accurate and rapid method for the determination of *M. maydis* soil inoculum level.

On the other hand, *M. maydis* can be isolated and identified from symptomatic plant tissues using microscopic morphological characteristics, but this method requires taxonomic expertise and time investment. Additionally, as noted earlier by Saleh et al. (2003) [25], *M. maydis* recovery from plant material is challenging. This difficulty is true even with heavily inoculated plant tissues, due to the pathogen’s slow development and the relatively high prevalence of other, more fast-growing fungi, specifically *Fusarium* spp. Indeed, late wilt disease is often accompanied by secondary plant pathogenic fungi contamination, enhancing the stem symptoms. Such pathogenic fungi opportunists are *Fusarium verticillioides,* causing stalk rot [15], and *Macrophomina phaseolina,* causing charcoal rot [17]. We invested considerable efforts in *M. maydis* isolation from infested commercial fields with a long history of late wilt harsh damages, but with minor success [26]. The use of Hygromycin-containing growth-media in moderate concentration (up to 50 µg/mL), which allows for *M. maydis* development, partly facilitated this task.

The presence of species-specific PCR primers enables the differentiation of *M. maydis* from other species in the *Gaeumannomyces-Harpophora* group [25]. These primers were used as a molecular diagnostic method to track disease progression in the tissues of maize plants grown in an infested field in northern Israel [15,27] and to study the interactions of *M. maydis* with its host [28]. The molecular assay, together with symptoms evaluation, was used to test chosen fungicides’ ability to restrain the pathogen [8,29]. This approach was applied, in vitro, in a series of experiments, comprised of a culture plate screening, followed by a detached root pathogenicity inoculation, and eventually a seedling assay [4].

The traditional PCR-based method is only capable of exposing changes in the amount of fungal DNA inside the plant tissues, above certain threshold levels. For example, in field experiments, it can detect *M. maydis* DNA only from days 40–50 onwards [15]. In greenhouse sprouts experiments, the PCR was rarely able to detect the fungal DNA in the host tissues before 22 days after sowing (DAS) [4,8]. However, in most experiments, the molecular method failed to detect *M. maydis* DNA in host tissues, even 40 DAS, due to this technique’s sensitivity limitations.

The present work focuses on the application of the quantitative Real-Time PCR (qPCR) protocol for evaluating the potential use of the maize cultivar Megaton (Hazera Seeds Ltd., Berurim MP Shikmim, Israel) compared to other cultivars for the detection and estimation of soils infested with *M. maydis*. To this end, in vitro seeds and in vivo sprouts pathogenicity assays were conducted (in an incubator and growth chamber, respectively). Seed germination and early development, seedlings emergence, root and shoot wet biomass, and phenological stage were evaluated and compared to qPCR molecular tracking results to identify a possible correlation between symptoms development and pathogens DNA quantity.

## 2. Materials and Methods

### 2.1. Field Observation for Assessing Maize Cultivars’ Resistance/Susceptibility to Late Wilt Disease

A field observation for the estimation of maize cultivars’ resistance levels to late wilt disease was conducted in the spring and summer of 2017 in Kibbutz Amir maize field southern area (Mehogi 1 corn plot), in the Hula Valley (Upper Galilee, northern Israel). This field has been known to be late wilt infested for many years [8] and has been used since 2013 as an experimental field to examine new maize cultivars’ resistance/sensitivity to late wilt disease.

The experimental field average air temperature during the growth period (May 25–August 28, 2017) was 27.6°C, with a minimum temperature of 11.3°C and a maximum of 41.3°C. The average soil temperature at the top 5 cm was 35.1 °C, with a minimum temperature of 20.4 °C and a maximum of 50.4 °C. Average humidity was 51.3% (with a minimum of 13.2% and a maximum of 85.5%). During this period, no precipitation was measured (data according to the Israel Northern Research and Development meteorological station). The experiment tested 12 fodder and six sweet maize cultivars (Table 1).

Plots, each containing one row, were arranged in the field using a randomized complete block design. A standard row spacing of 96.5 cm was used. Each row was 12 m long and included 8 maize plants·m^−1^ (approximately 96 plants per row). The field was watered with a 16 mm drip line with 50 cm drip spacing (Dripnet PC1613 F, Netafim, Tel Aviv-Yafo, Israel) applied for two rows. The drip flow rate was 2 L/h, and the field was irrigated during the overall growth season, with a total of 450 mm. All of the plants received insecticides and fertilization according to a growth protocol recommended by the Israel Ministry of Agriculture and Rural Development, Consultation Service (Shaham, Beit-Dagan, Israel). Sowing conducted on May 25, 2017, and germination occurred one day later by watering the field with a frontal irrigation system. The fertilization stage occurred on July 20, 2017 (56 DAS), and the fruit ripening stage in most cultivars was set 14 days later on August 13, 2017 (80 DAS).

The dehydration degree was evaluated 66, 80, 84, and 95 DAS. Calculating the percentage of plants with typical maize late wilt dehydration symptoms was performed according to the upper leaves’ color change to light-silver and then to light-brown. A plant was considered dried when it shows harsh drought symptoms—over 50% of its parts are dehydrated. Classification of the maize cultivars according to their degree of late wilt disease sensitivity was performed using four criteria (previously determined by Dr. Tsafrir Weinberg), as described in Table 2. Later, on the harvest day of sweet maize cultivars (78 DAS), the levels of *M. maydis* DNA were measured using qPCR analysis for all experiment cultivars, as will be detailed below. For each cultivar, the qPCR analysis was conducted on three representative replications (plants). The fodder maize cultivars were harvested 95 days post-sowing.

### 2.2. Fungal Isolates and Growth Conditions

One selected isolate of *M. maydis*, *Hm2* (CBS 133,165), was used for this work. This isolate is a pure monosporic, pathogenic strain validated by its morphological, microscopic, and molecular characteristics [15,25]. The *Hm2* strain is kept at the CBS-KNAW Fungal Biodiversity Center, Utrecht, The Netherlands. All fungal colonies were grown routinely on potato dextrose agar (PDA) (Difco, Detroit, MI, USA) in complete darkness at 28 ± 1 °C.

### 2.3. In Vitro Seed Infection

To examine the pathogens’ ability to infect maize seeds in vitro, seeds were inoculated with the fungus and then tested for the fungus’ DNA presence in the inner tissues in a previously developed method [8]. The sweet maize hybrids, Megaton cv., Prelude cv. (from SRS Snowy-River seeds, Australia, supplied by Green 2000 Ltd., Bitan Aharon, Israel), or Royalty cv. (from Pop Vriend Seeds B.V., Andijk, The Netherlands, provided by Eden Seeds, Reut, Israel) were chosen for the seed pathogenicity trial. The Megaton cv. is a highly susceptible maize hybrid studied here in the context of late wilt disease for the first time. The other two cultivars, Prelude (late wilt disease high susceptible cv.) and Royalty (moderately resistant cv.), are representative maize hybrids well studied in the lab [25] and the field [4,8,15,29]. 

Ten seeds of each selected maize genotype were dipped in 1% (*v*/*v*) sodium hypochlorite for 3 min, thoroughly washed in autoclaved double-distilled water (DDW), and then maintained with 20 mL sterile DDW in a 250 mL Erlenmeyer flask. Three 6-mm-diameter *M. maydis* colony agar disks were added to each Erlenmeyer flask. These agar disks were cut previously from *M. maydis* colonies margins. These colonies grew on PDA in the dark at 28 °C for about five days. The Erlenmeyer flasks were maintained at 28 °C in the dark in a rotary shaker at 150 revolutions per minute (RPM) for one week. At the end of the experiment, determination of seeds germination percentage, wet biomass of all the seeds in each treatment, and wet biomass of the germinated seeds was conducted using analytical scales. A germinating seed was defined as a seed in which the radicle scored the seed coat. Each treatment and the non-inoculated control (seeds soaked only in water) comprised six independent replications. The whole experiment was repeated twice, and similar results were obtained. The molecular detection at the experiment end was conducted on samples of three seeds from each repeat for every treatment. The samples were disinfected with an ethanol suspension (70%) and then washed with sterile DDW to ensure no fungus hyphae are attached to the seeds’ surface. The seeds were ground to a powder in liquid nitrogen, and DNA was extracted and used as a template for the qPCR reactions, as will be detailed below.

### 2.4. Growth Chamber Seedlings Assay

Maize sprout susceptibility assays were performed under controlled conditions and aimed at achieving two goals. The first goal was to identify variations in the severity of the symptoms of plants cultivated on naturally infested field soils taken from two different locations. Soil samples were taken from commercial fields, having a long history of *M. maydis* contamination: the Kibbutz Amir field soil [8,29] and the Kibbutz Neot Mordechai field soil [15,27]. The direct examination of those soils impact on late wilt disease symptoms progression was carried out without the enrichment with the pathogen (added in the other growth chamber experiment). Additionally, the symptoms evaluation was performed by studying the differences among selected maize cultivars and between those cultivars and *S. viridis* (green foxtail), a recently discovered new host of this fungus [18]. This is important to identify the most appropriate test plant for the soil assay. Our second aim was to use the preferable cultivar and determine the optimal growth period duration that will be used to inspect susceptibility to late wilt infested soils.

Each experimental group comprised of six biological replicates (pots). Each pot was sown with five seeds. Maize hybrids Megaton cv. and Royalty cv., and *S. viridis* were inoculated with *M. maydis* as previously described [25]. Briefly, seeding was performed in a two-liter pot about 4 cm beneath the ground surface. The soil was naturally infested peat soil from the Kibbutz Amir maize field (Hula Valley, Upper Galilee, northern Israel), which has been *M. maydis* infested for many years, mixed with 30% Perlite No. 4 (for aerating the soil(. The negative control (uninfected soil) in the pot experiments was taken from a commercial field alongside soil that had no history of late wilt disease. If *M. maydis* infestation did present in the negative control soil, it was estimated to be very low. Watering was performed by adding 100 ± 10 mL tap water every 72 h. All the plants were raised in a controlled condition in a growth chamber at a temperature of 25 ± 2 °C, a relative humidity of 30% and a 16-h photoperiod illuminated by cool-white fluorescent tubes (Philips, Eindhoven, The Netherlands).

To ensure as high and uniform a disease pressure as possible, the inoculum method comprised two steps. First, 20 g sterilized infected wheat seeds were added to each pot with the seeding. These seeds had been incubated previously for three weeks in the dark at 28 °C with 10 *M. maydis* colony agar disks (per 100 g seeds). These sterilized infected seeds were used to scatter the pathogen in the soil, as previously described [15,30]. Second, according to [4], two culture agar disks (6-mm-diameter) from five-day-old *M. maydis* colonies (grown in the dark at 28 °C) were added to each plant’s upper roots (4 cm beneath the surface) four days after seeding (with the emergence of the plants above the ground surface).

At three intervals at the age of 10, 20, and 30 days post-sowing, the seedlings were examined for phenological development and plant height. Also, roots and above-ground fresh-weight parts were measured. At the above time points, the seedlings were removed gently from the ground, rinsed thoroughly under running tap water and then dried softly with paper towels. The root and above-ground parts of each seedling were segregated by a scalpel, and the wet biomass was measured separately using analytical scales. The above-ground parts height (from the first node to the shoot tip) of each sprout was measured individually. The whole experiment was repeated twice, and similar results were obtained.

For DNA extraction, plants were washed twice in DDW for 30 s. Tissue testing was achieved by cutting a cross-section of about 2 cm in length from the root’s uppermost part and the near-surface hypocotyl of each plant. Five plants from each pot were sampled, the samples were combined, and the total fresh weight was adjusted to 0.7 g and regarded to be one replicate. DNA was extracted and used as a template for the qPCR reactions, as will be detailed below.

### 2.5. DNA Isolation and Extraction

Tissue samples were moved to universal extraction bags (Bioreba, Reinach, Switzerland) and 4 mL CTAB buffer (0.7 M NaCl, 1% cetyltriammonium bromide, 50 mM Tris- hydrochloric acid pH 8.8, 10 mM EDTA and 1% 2-mercaptoethanol) were added to each bag. The samples were ground with a hand tissue homogenizer (Bioreba, Reinach, Switzerland) for 5 min until the samples were fully homogenous. The homogenized tissues underwent DNA extraction, according to [31], with minor modifications [25]. After grinding the tissue samples, 1.2 mL from this mixture was kept for 20 min at 65 °C. The samples were centrifuged for 5 min at 25 °C at 14,000 rpm. The upper lysate part (usually 700 µL) was extracted with an equal volume of chloroform/isoamyl alcohol (24:1). Following mixing by vortex, the mixture was repeated twice. The supernatant (approximately 300 µL) was then moved to a new Eppendorf tube and mixed with cold isopropanol (2:3). The DNA suspension was mixed gently by inverting the tube several times, maintained for 20–60 min at 20 °C, and centrifuged (14,000 rpm for 20 min at 4 °C). The precipitated DNA isolated was resuspended in 0.5 mL 70% ethanol. After additional centrifugation (14,000 rpm at 4 °C for 10 min), the DNA pellet was dried in a sterile hood overnight. The DNA was suspended by the addition of 100 µL of HPLC-grade water and maintained at −20 °C until use in the qPCR reactions.

### 2.6. Molecular Diagnosis

The molecular method used was recently described in detail [17]. Briefly, three representative plants from each treatment were selected arbitrarily for qPCR analysis. Each biological replicate was tested in triplicate using qPCR to confirm the reliability of the results. The qPCR reactions were executed using the ABI PRISM^®^ 7900 HT Sequence Detection System (Applied Biosystems, Foster City, CA, USA) for 384-well plates. A reaction volume of 5 µL (in total) contained 2 µL of the DNA extract sample, 2.5 µL of the Universal SYBR^®^ Green Supermix iTaq™ (Bio-Rad Laboratories Ltd., Rishon Le Zion, Israel), 0.25 µL of each of the forward and reverse primers (at a concentration of 10 µM of each primer per well). The qPCR plan was: 1 min at 95 °C precycle activation stage, 40 cycles of denaturation (15 s at 95 °C), annealing and extension (30 s at 60 °C), and eventually a melting curve analysis. The target A200a, *M. maydis*-specific DNA, was evaluated against a reference “housekeeping” gene—the mitochondria-cytochrome c oxidase, COXI gene (sequences in [32]). This reference gene encoding the eukaryotic mitochondria respiratory electron transport chain’s last enzyme was used to normalize the amount of DNA. Calculating the relative DNA abundance was according to the ΔCt model. The same efficacy was assumed for all samples. All amplifications were performed in triplicate.

### 2.7. Statistical Analyses

In all experiments, a completely randomized statistical design was used. Data analysis and statistics were fulfilled using the JMP program, 7th edition, SAS Institute Inc., Cary, NC, USA. The Student’s t-test was used for the analysis of *M. maydis* infection results (with a significance threshold of *p* = 0.05) and for comparisons of treatment means to the control. In the field trials’ qPCR-based-molecular DNA tracking, there is an objective difficulty to achieve uniform repeats, due to changes in environmental conditions, and the non-uniform spreading nature of the late wilt disease pathogen [32]. Consequently, this resulted in relatively high standard error values, and in most of those tests, no statistically significant difference could be measured in comparison to the control. This is also true for indoor growth experiments, whereas achieving a uniform infection with this particular pathogen Israeli strains is challenging [8].

## 3. Results

Annual field assessments of new maize varieties for the determination of their degree of resistance to late wilt disease have been conducted by the Consultation Service (Shaham, Beit-Dagan), Israel Ministry of Agriculture and Rural Development, and by the Israel Northern R&D for over a decade. In this procedure inspection, each cultivar was tested over three growth seasons (years) in order to provide a trustworthy recommendation to farmers as to whether to grow a specific maize cultivar on *M. maydis*-infested soil. During this routine evaluation in 2017, a newly developed hybrid, Megaton cv., was discovered to have late wilt disease hyper-susceptibility (Figure 1).

At 66 days post-seeding (10 days after fertilization, DAF), the new Megaton cv. was dried entirely (Figure 1), with total yield loss (Table 3). At the same time, most of the other cultivars tested showed some degree of resistance and had a healthy appearance with or without minor dehydration symptoms.

Tracking the *M. maydis* fungal DNA within the Megaton cv. root tissues (at 66 DAS) supported the severe above-ground symptoms with over 120-fold higher relative *M. maydis* DNA levels compared to the nearest most sensitive sweet maize cultivar (Royalty cv., Figure 2). The pathogen spreading within the Megaton cultivar’s root system was prominently higher also in comparison to the most late-wilt-sensitive fodder maize cultivar tested this year, the 2572 cv. Overall, *M. maydis* DNA fluctuation among the different maize varieties (Figure 2) well reflects the degree of their sensitivity or resistance to the disease expressed in their dehydration symptoms (Table 3).

A dedicated set of experiments aimed at adjusting the seedling assay under controlled conditions will allow us to make a relatively rapid determination of the degree of soil infestation with the late wilt causal agent. To this end, we used the newly discovered Megaton cv. as a potential check genotype and measured its disease severity in response to the soils’ prevalence of the pathogen (Figure 3). Indeed, two soil samples from commercial fields having a long history of *M. maydis* contamination, the Kibbutz Amir field soil [8,29], and the Kibbutz Neot Mordechai field soil [15,27] evoked a noticeable response. Planting the highly susceptible Megaton cv. in those field soils caused an inhibitory effect on the plants’ growth parameters after 30 days of growth in a controlled environment growth chamber (Figure 3).

However, these measurements can sometimes be inconsistent. For example, the Kibbutz Amir soil led to significant (*p* < 0.05) inhibition of the development of the plant’s above-ground parts (phenological state, plant height and shoot fresh weight), while an opposite picture was revealed in the Kibbutz Neot soil (in most measures, except for plant height). Interestingly, the roots of the inspected Megaton cv. reacted differently from the shoot. Plants in the Kibbutz Amir infested soil had a more extensive root system expressed in wet weight (compared to the control) in contrast to those grown in the Kibbutz Neot soil, which had a reduced root wet weight. Thus, relying only on growth parameters is not sufficient for soil contamination evaluation.

To overcome this limitation and to achieve consistent and accurate evaluation of the degree of soil infestation, the sensitive qPCR molecular tool provides an excellent alternative. The qPCR assay consistently detected *M. maydis* DNA from all sample types except for the negative controls: uninfected maize tissue. The use of this technique for the soil bio-assay will be demonstrated in the following experiments.

A sequential experiment was designed to evaluate Megaton cv. late wilt sensitivity in comparison to two other well-studied [25] representative sweet maize cultivars, Prelude cv. (late-wilt-sensitive hybrid) and Royalty cv. (moderately resistant hybrid, Figure 4). In this setting, in a seed assay in Erlenmeyer flasks, the Megaton cv. was severely affected by the *M. maydis* after one week. While the Prelude cv. and Royalty cv. seeds had a moderate response to the pathogen, as reflected in 12–15% germination inhibition and 3–17% germinate seeds biomass reduction, in the Megaton cv., these parameters were significantly higher (*p* < 0.05) and exceeded 30% (Figure 4).

The qPCR analysis results (Figure 5) were in line with the seeds’ germination results. The infected Megaton cv. seeds had 15.8 relative *M. maydis* DNA levels, 6.2 times higher than those of the Royalty cv., and 1,576 times higher than those of the Prelude cv.

The following step was to inspect Megaton cv. susceptibility in potted sprouts up to the age of 40 days. Here, the Megaton cv. was compared to the Royalty cv., and plant development measures were documented. At 7 DAS, the emergence of seedlings above the ground surface revealed the late wilt response (Figure 6). Both cultivars suffered from growth inhibition as a result of the pathogen presence. The above-ground emergence inhibition reached 23% of the seeds in the Megaton cv. and 45% in the Royalty cv. (Figure 6A). Moreover, not only the number of emerging sprouts was affected, but also the number of leaves and their size, as can be seen in the experiment’s photo (Figure 6B).

Indeed, close inspection of the growth parameters at 10, 20, and 30 DAS revealed that the phenological state, the above-ground parts fresh weight and height, and the weight of the roots were all inhibited by the *M. maydis* presence (Figure 7). At 10 DAS, the Royalty cv. symptoms were more severe than the Megaton cv. symptoms. However, at 20 DAS, and even more at 30 DAS, the Megaton cv. exterior symptoms were significantly (*p* < 0.05) higher, with about a 30% delay in phenological development and plan height growth, and 40% and 70% reduction in shoot and root fresh-weight, respectively. Interestingly, the Royalty cv., which is considered to be a moderately resistant cultivar [25], showed prominent symptoms at 10 DAS. Still, those symptoms gradually disappeared during the growth period, and eventually, only a minor plant height and root weight reduction were measured (Figure 7 and Figure 8A).

The molecular tracking of *M. maydis* DNA in the roots (Figure 8B,C) of the experiment described above showed an expected elevation in the Megaton cv. plants. The relative DNA levels in those plants increased from 0.45 at 20 DAS to 5.48 at 30 DAS. The DNA levels were 173 and 945 times higher than the pathogen DNA levels in the Royalty cv.’s roots, respectively. We compared the maize plants results to the pathogen DNA variations within the roots of *S. viridis* (green foxtail), a recently discovered new host of this fungus [18]. In the *S. viridis* plants that were grown under the same conditions during the same periods, *M. maydis* DNA levels were similar to those in the maize Royalty cv. (Figure 8B,C).

## 4. Discussion

There is an urgent need to develop a more rapid, accurate, and specific assay to determine the degree of *M. maydis* infestation in suspicious soils. Today, no such method exists, and relying on traditional media plant protocols to isolate the pathogen from the ground has several disadvantages that make this solution impractical. A limitation of this technique includes the difficulty in isolating the pathogen from the soil due to its scattered nature and low prevalence. Additionally, the pathogen has a relatively slow growth rate of about 1 cm per day on PDA media plates and optimal temperature [15], and it is considered a poor competitor in a mixture of microorganisms. Thus, from the soil mycoflora existing in the sample, rapid-growth fungi such as *Fusarium* spp. will most probably take over and cover the plate [33]. This limitation is true even if antibiotics are added to the medium [25]. Hence, the media plate technique to isolate the pathogen directly from the soil is time-consuming and produces inconsistent results.

The seedling pathogenicity inspection is a preferable method for soil testing. It provides more reliable and relatively rapid results while minimizing the risk of missing *M. maydis* identification (receiving false-negative results). To this end, we developed and validated a qPCR-based soil bioassay that facilitates detecting and tracking the pathogen in a testing maize plant, the Megaton cv. This Megaton cv. has a high susceptibility to late wilt disease and was discovered during the routine annual inspection of new cultivars conducted in an *M. maydis*-infested commercial field. Similarly, other highly sensitive maize hybrids could be used for the same purpose. The finding that Megaton is so susceptible in the field provides encouragement to screen diverse inbreeds by the field screening method, which may lead to an identified more ideal trap plants that would allow pathogen detection reproducibly within days of planting.

Interestingly, in the results presented here, the moderate resistance Royalty cv. was relatively more susceptible to fungal infection then the Megaton cv., at the earliest stages of infection, both 7 DAS and 10 DAS (Figure 6 and Figure 7, respectively). One possibility for these results is that Royalty cv. has some sort of adult plant resistance that starts acquiring and manifesting after two weeks of growth. It would have been helpful to have conducted a DNA analysis at these earlier stages, but the ability of the qPCR-based DNA analysis to detect the pathogen DNA in potted sprouts before 20 DAS is limited [25].

The advantage of the proposed bioassay using a susceptible cultivar and a qPCR-based method is its ability to detect the fungus even if it scattered in the soil. This assay enables us to isolate and enrich the maize late wilt pathogen from the soil using a trap plant. The use of this bioassay is essential for study *M. maydis* distribution and to provide an estimation of its infestation degree in commercial fields. Such data are necessary for decision-making among growers to reduce disease damage and provide risk assessment. The soil bioassay data can provide insurance companies with vital information that will help them to support the farmers. Identifying the fungus will allow the farmer to make decisions about sowing time, choosing the appropriate maize cultivar for planting, the application of disease management strategies, the implementation of a prevention plan, and more. All these means may also help quarantine the infected areas and prevent the disease from spreading to new fields. Together with these advantages, it should be noted that the sensitivity of the bioassay method presented in this work was not specified. It would be beneficial to determine the minimum concentration of the pathogen in the soil that would allow the detection of the pathogen in the plant. This should be the focus of a subsequent study. Additionally, the objective difficulty of achieving a uniform infection with this particular pathogen Israeli strains may be reduced if the number of plants used in each assay is increased.

The proposed assay may also be used to evaluate the virulence of different *M. maydis* isolates. Indeed, *M. maydis* strains differ in morphology and mode of infection [34]. For example, four different lines of *M. maydis* isolated in Egypt showed different abilities to establish and infect corn plants [30,33,35]. In southern Portugal and Spain, an analysis of 14 *M. maydis* isolates was performed using 32 different maize varieties [36]. One of the isolates was extremely virulent and caused intense symptoms that included significant decreases in the fresh weight of both above-ground parts and roots.

Such extremely virulent isolates are an alarming threat to resistant maize cultivars as well. Indeed, *M. maydis* may also spread in the tissue of resistant corn genotypes without causing any visible symptoms [15], and the seeds of the same resistant cultivar may spread the disease. This hints at the possibility that the pathogen acts as an endophyte in resistant maize cultivars, and it may become an opportunist, causing disease under certain circumstances, or it may undergo pathogenic variations and become aggressive.

A TaqMan probe assay was developed recently to target *M. maydis* [21]. This newly developed methodology was demonstrated in a field experiment through the screening of potentially infected maize roots, revealing a high specificity and proving to be a suitable tool to ascertain *M. maydis* infection in maize. Its high sensitivity makes it very efficient for the early diagnosis of the diseases and also for certification purposes. It will be most interesting to study the combination of the Megaton cv. pot bioassay together with the TaqMan probe assay in order to achieve an improved soil assay with maximum efficiency and sensitivity.

## 5. Conclusions

The fungus *Magnaporthiopsis maydis* is a dangerous pathogen, destructively affecting sensitive maize plants at the maturity stage by causing rapid and sudden wilting. This work is part of a continuous scientific effort and advances our ability to prevent the late wilt disease spread and to control it. To face this challenge, we present a new soil bioassay that could sensitively detect *M. maydis* infestation and study its prevalence. The assay is based on the cultivation of a new late wilt hyper-susceptible maize plant (Megaton cv.) on the inspected soils and thus ease the isolation and enrichment of the pathogen from the soil. The novel method also includes a highly sensitive detection approach based on a recently developed qPCR method. The use of this bioassay is essential for studying *M. maydis* distribution and for providing an estimation of its infestation degree in commercial fields. The data received from this assay enabling decision-making for the growers that would reduce disease damage, and risk assessment for the insurance companies that would allow them to support the farmers. All these means may also help quarantine the infected areas and prevent the disease from spreading to new fields. The proposed assay may also be used to evaluate the virulence of different *M. maydis* isolates.

## Figures and Tables

**Figure 1 jof-06-00107-f001:**
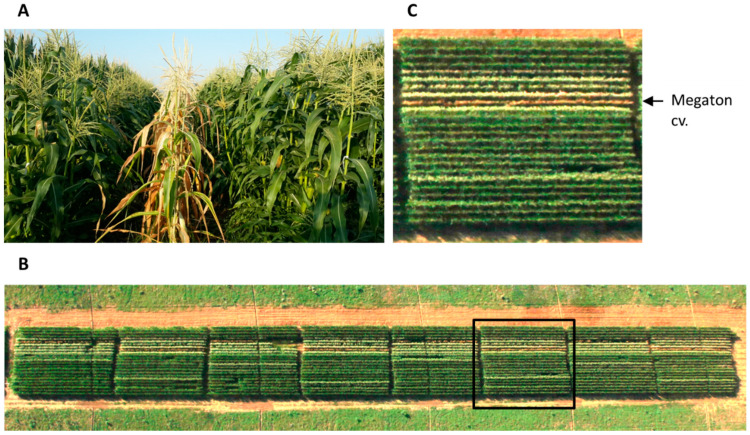
Field observation for assessing maize cultivars’ resistance/susceptibility to late wilt disease. The commercial maize field near Kibbutz Amir in the Hula Valley (Upper Galilee, northern Israel) was photographed 66 days after sowing (DAS), 10 days after fertilization (DAF, at July 30, 2017). (**A**). The wilted Megaton cv. plot. (**B**). An aerial photograph of the test field (photographed by Asaf Solomon). (**C**). Magnification of a portion marked by a black box in the whole field photo (**B**). The wilted Megaton cv. can be seen from the air as a brown strip.

**Figure 2 jof-06-00107-f002:**
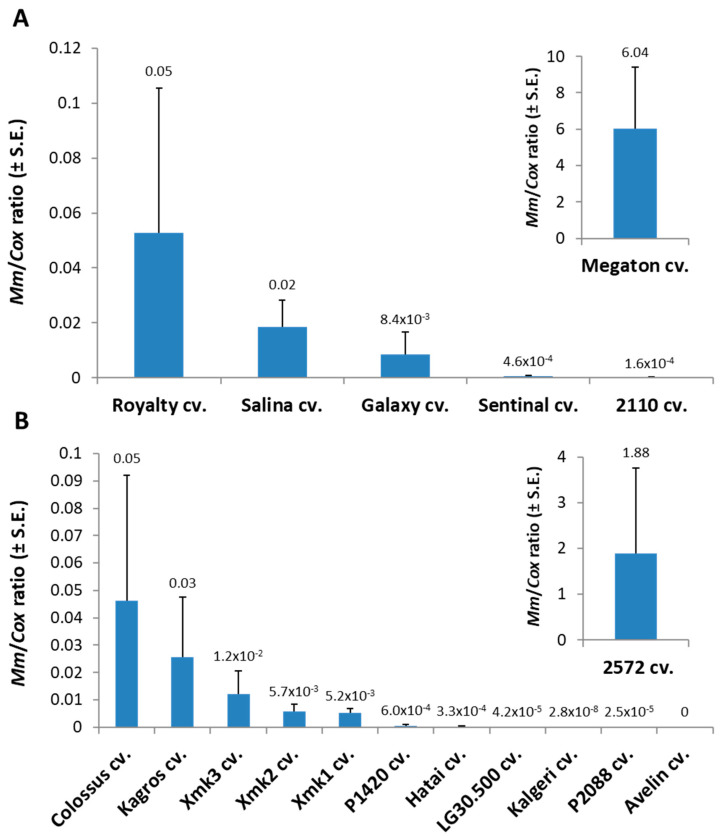
Quantitative real-time PCR (qPCR) detection of *Magnaporthiopsis maydis* in the field-tested maize cultivars (presented in Figure 1). (**A**) Sweet maize cultivars. (**B**) Fodder maize cultivars. On the harvest day of the sweet maize cultivars (78 DAS), the levels of *M. maydis* DNA were measured by qPCR analysis for all experiment cultivars. The maize cultivars tested are detailed in Table 3. Tissue samples were from the root’s topmost part. The *y*-axis parameters are *M. maydis* relative DNA (*Mm*) levels normalized to the cytochrome c oxidase (*Cox*) DNA. Upper vertical bars represent the standard error of the mean of three replications (plants).

**Figure 3 jof-06-00107-f003:**
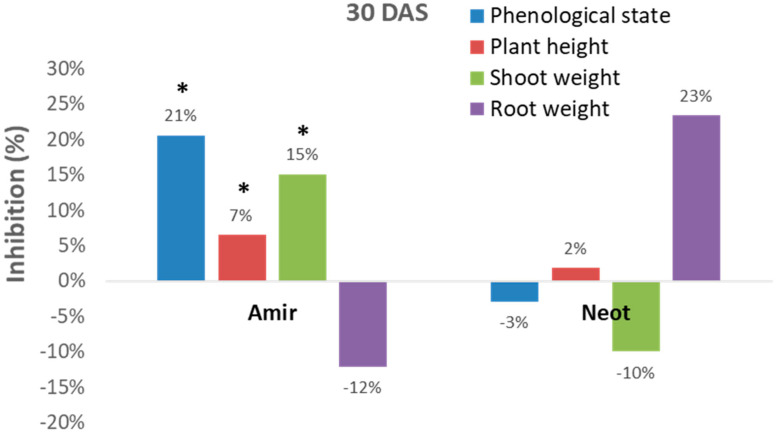
Soil bioassay using Megaton cv. as a testing plant. The late-wilt-sensitive maize Megaton cv. was sown in two commercial soils that have a long history of late wilt disease—the Kibbutz Amir field and the Kibbutz Neot Mordechai field (both located in the Hula Valley, Upper Galilee, northern Israel). The experiment was conducted in six replications in pots in a growth chamber under controlled conditions. The plants’ growth parameters were evaluated 30 DAS. Values represent the average difference (% inhibition) of the plants that grew on naturally infested soil, compared to the control. Control group comprises maize plants grown on soil taken from a field nearby, that has minor levels of *M. maydis* infestation. Weight—wet biomass. Vertical upper bars signify the standard error of the mean of six repeats (pots, each having five plants). Significance difference from the control is shown as * = *p* < 0.05.

**Figure 4 jof-06-00107-f004:**
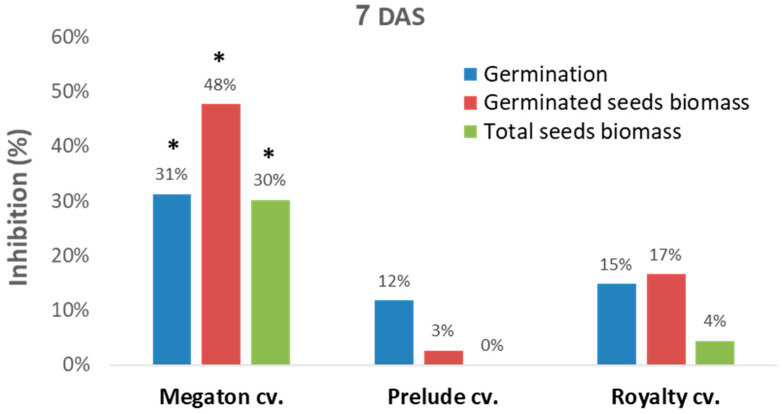
In vitro seed assay. The Megaton cv. was evaluated in comparison to two other representative sweet maize cultivars, Prelude cv. (late-wilt-sensitive hybrid) and Royalty cv. (moderately resistant hybrid). The seed assay was conducted with six replications in Erlenmeyer flasks (each containing 10 seeds) under controlled conditions. The germinated seed percentages and wet biomass were evaluated 7 DAS. Values represent the mean difference (% inhibition) from the non-inoculated control groups. Significance difference from the control is shown as * = *p* < 0.05.

**Figure 5 jof-06-00107-f005:**
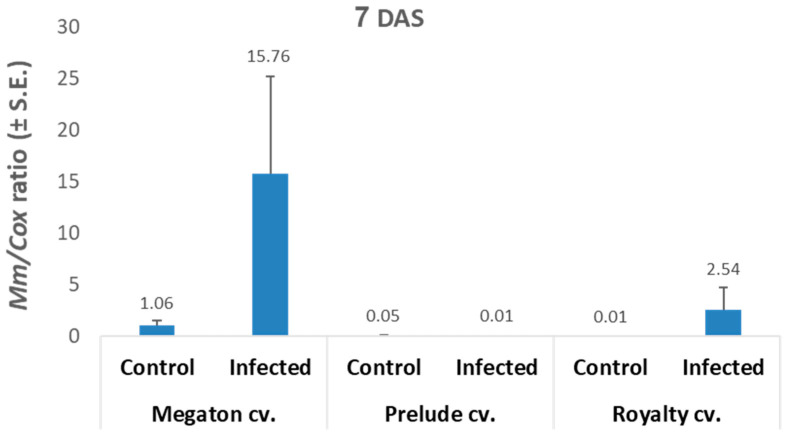
In vitro seed assay, qPCR analysis. The qPCR-based molecular method was used at 7 DAS to identify *M. maydis* DNA levels in the maize seeds tested in the experiment described in Figure 4. Upper vertical bars represent the standard error of the mean of six replications (representative plants of each cultivar). The target *M. maydis*-specific DNA was evaluated against a reference “housekeeping” gene—the mitochondria-cytochrome c oxidase, *COXI* gene.

**Figure 6 jof-06-00107-f006:**
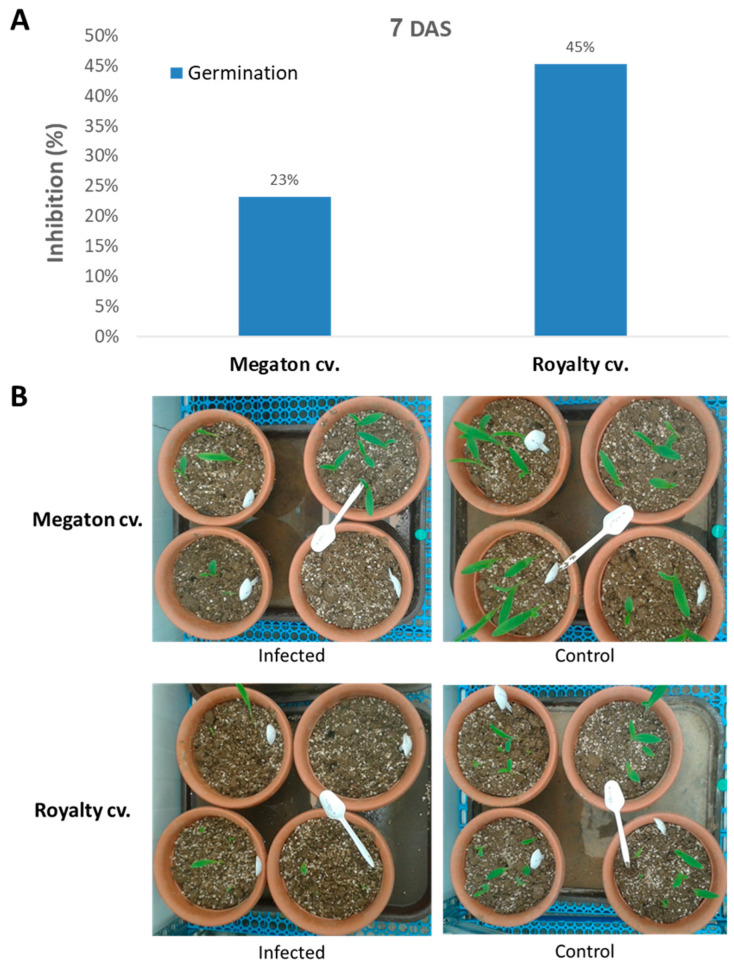
Emerging seedlings pathogenicity assay. The late-wilt-hypersensitive maize Megaton cv. germination and first development stages were appraised in comparison to Royalty cv. (moderately resistant hybrid). The experiment was conducted in six replications (pots, each containing five plants) in the same setting and with the same control as in Figure 3. (**A**) The above-ground emergence inhibition was evaluated 7 DAS. Values represent the mean difference (% inhibition) from the non-infected control group. (**B**) Photograph of representative pots of each of the experiment groups, at the same age.

**Figure 7 jof-06-00107-f007:**
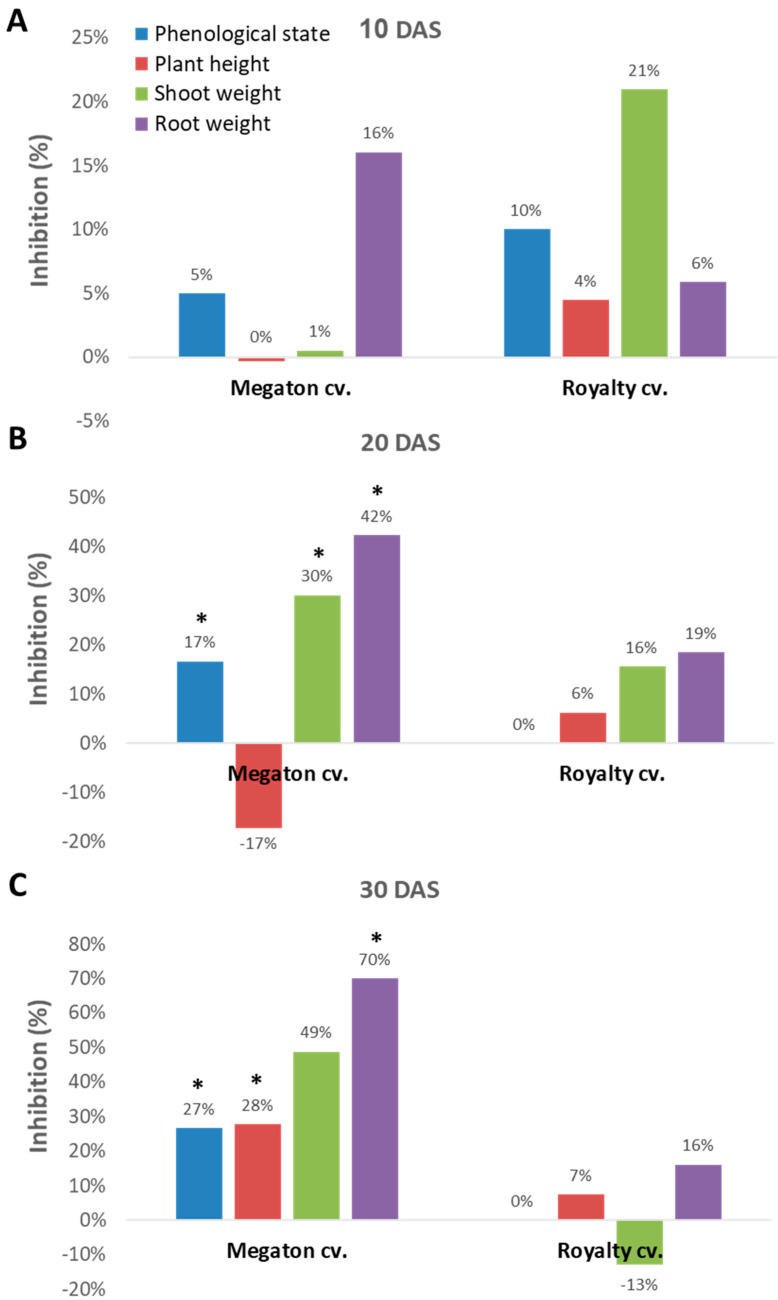
Seedling pathogenicity assay: growth parameters at 40 DAS. The growth parameters of the plants in the experiment presented in Figure 6 were evaluated 10 (**A**), 20 (**B**), and 30 (**C**) days after sowing. These parameters include the phenological development stage, the roots, and the above-ground fresh weight and plant height. Values represent the mean difference (% inhibition) of six repeats from the control group. Significance difference from the control is showed as * = *p* < 0.05.

**Figure 8 jof-06-00107-f008:**
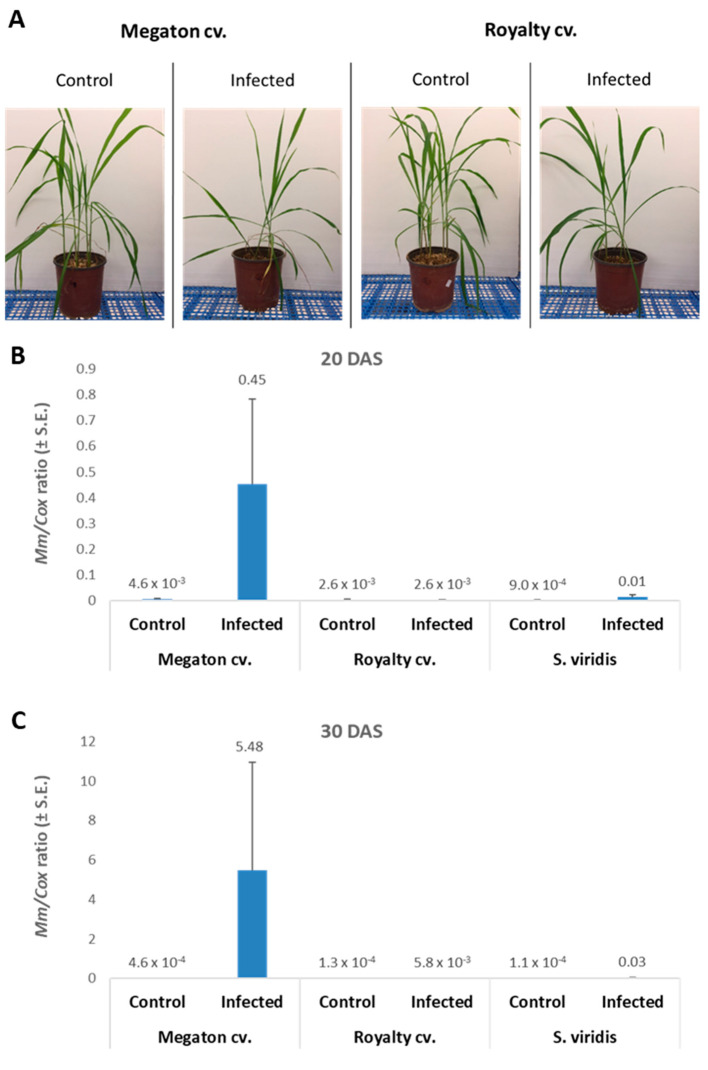
Seedling pathogenicity assay: qPCR analysis. **A**. Pictures of representative plants from the experiment described in Figure 6 were taken 30 DAS. The plants were tested for the presence of *M. maydis* DNA inside their root tissues 20 (**B**) and 30 (**C**) post-sowing. The maize varieties (Megaton cv. and Royalty cv.) evaluation was compared to the pathogen DNA variations within the roots of *Setaria viridis* (green foxtail), a recently discovered new host of this fungus [18]. The *S. viridis* plants were grown under the same condition for the same periods as the maize cultivars. The *y*-axis parameters are *M. maydis* relative DNA (*Mm*) levels normalized to the cytochrome c oxidase (*Cox*) DNA. Vertical upper bars signify the standard error of the mean of three repeats (plants).

**Table 1 jof-06-00107-t001:** Maize cultivars tested for late wilt disease sensitivity/resistance.

Number	Cultivar	Type	Seed Company	Supply Company
1	2572	Fodder	KWS, Einbeck, Lower Saxony, Germany	CTS Group, Tel Aviv, Israel
2	Kagros	Fodder	KWS, Einbeck, Lower Saxony, Germany	CTS Group, Tel Aviv, Israel
3	Colossus	Fodder	KWS, Einbeck, Lower Saxony, Germany	CTS Group, Tel Aviv, Israel
4	Xmk2	Fodder	Hungaroseed, Budapest, Hungary	Gambit seeds, Rosh Pinna, Israel
5	Xmk1	Fodder	Hungaroseed, Budapest, Hungary	Gambit seeds, Rosh Pinna, Israel
6	P1420	Fodder	Pioneer Hi-Bred International, Johnston, Iowa, United States	Gadot-Agro, Givat Brenner, Israel
7	Xmk3	Fodder	Hungaroseed, Budapest, Hungary	Gambit seeds, Rosh Pinna, Israel
8	P2088	Fodder	Pioneer Hi-Bred International, Johnston, Iowa, United States	Gadot-Agro, Givat Brenner, Israel
9	LG30.500	Fodder	Limagrain, Saint-Beauzire, Puy-de-Dôme, France	Hazera Seeds Ltd., Berurim MP Shikmim, Israel
10	Avelin	Fodder	Limagrain, Saint-Beauzire, Puy-de-Dôme, France	Hazera Seeds Ltd., Berurim MP Shikmim, Israel
11	Hatai	Fodder	Semillas Fitó, Barcelona, Spain	Tarsis Inc., Petach Tikva, Israel
12	Kalgeri	Fodder	Semillas Fitó, Barcelona, Spain	Tarsis Inc., Petach Tikva, Israel
13	Megaton	Sweet	Zeraim Gedera-Syngenta, Kibbutz Revadim, Israel	Hazera Seeds Ltd., Berurim MP Shikmim, Israel
14	Sentinal	Sweet	Hazera Seeds Ltd., Berurim MP Shikmim, Israel	Hazera Seeds Ltd., Berurim MP Shikmim, Israel
15	Salina	Sweet	Zeraim Gedera-Syngenta, Kibbutz Revadim, Israel	Zeraim Gedera-Syngenta, Kibbutz Revadim, Israel
16	Royalty	Sweet	Pop Vriend Seeds B.V., Andijk, The Netherlands	Eden Seeds, Reut, Israel
17	Galaxy	Sweet	Snowy-River Seeds, Victoria, Australia	Green 2000, Bitan Aharon, Israel
18	2110	Sweet	Zeraim Gedera-Syngenta, Kibbutz Revadim, Israel	Zeraim Gedera-Syngenta, Kibutz Revadim, Israel

**Table 2 jof-06-00107-t002:** Classification of maize cultivars according to late wilt disease susceptibility.

Degree of Late Wilt Disease Susceptibility ^1^	Percentage of Dehydrated Plants		
	Before the Fruit Ripening	During the Fruit Ripening	After the Fruit Ripening
Highly resistant	0%	0%	0–5%
Minor symptoms	0%	0–5%	5–10%
Partly resistant	0–5%	5–10%	10–25%
Sensitive	5–10%	10–100%	25–100%

^1^ Colors represent the degree of the cultivars susceptibility to late wilt disease.

**Table 3 jof-06-00107-t003:** The 2017 maize cultivars field examined for late wilt disease sensitivity/resistance. ^1.^

Cultivar	Type	Percentage of Dehydrated Plants				Degree of Late Wilt Disease Sensitivity ^3^
		66 DAS ^1^	80 DAS	84 DAS ^2^	95 DAS	
2572	Fodder	0	43.75	58.88	100	Sensitive
Kagros	Fodder	0.13	38.75	54.38	100	Sensitive
Colossus	Fodder	0	15.00	14.63	96.25	Sensitive
Xmk2	Fodder	0	15.00	24.13	96.25	Sensitive
Xmk1	Fodder	0	8.75	7.75	88.38	Sensitive
P1420	Fodder	16.63	35.71	27.13	86.50	Sensitive
Xmk3	Fodder	0	4.38	18.25	82.88	Partly resistant
P2088	Fodder	0	7.50	4.38	68.38	Highly resistant
LG30.500	Fodder	0	13.13	5.88	60.63	Highly resistant
Avelin	Fodder	0	0	1.38	31.63	Highly resistant
Hatai	Fodder	0	1.43	8.88	31.38	Highly resistant
Kalgeri	Fodder	0	0	0.63	12.75	Highly resistant
Megaton	Sweet	100	-	-	-	Hypersensitive
Sentinal	Sweet	2.14	-	-	-	Minor symptoms
Salina	Sweet	0.13	-	-	-	Highly resistant
Royalty	Sweet	0	-	-	-	Highly resistant
Galaxy	Sweet	0	-	-	-	Highly resistant
2110	Sweet	0	-	-	-	Highly resistant

^1^ At 66 days after sowing (DAS), all of the sweet maize cultivars were at the pre-ripening stage. The sweet maize cultivars were harvested at 78 DAS. ^2^ At 84 DAS; all of the fodder cultivars, except Hatai and Kaleri, reached the ripening stage. ^3^ Colors represent the degree of the cultivars susceptibility to late wilt disease, described in Table 2.

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
