# Peer review of "Soil Bioassay for Detecting Magnaporthiopsis maydis Infestation Using a Hyper Susceptible Maize Hybrid"

_jof, 2020, doi:10.3390/jof6030107_

Round 1
Reviewer 1 Report
As I had said before, the manuscript is very interesting, focusing one of the majors contains to maize production in several countries. My major concern with the first version of the manuscript was regarding the experimental design of the experiments, that was now clarified by the authors.
This version is clearly improved from the last version, the authors made a good effort and considered the reviewers comments. In my opinion the manuscript can now be accepted for publication.
Author Response
We thank the reviewer for the essential and helpful corrections, suggestions, and advice. We are sure that this contribution significantly improved the manuscript. Thank you.
Reviewer 2 Report
The authors did a good job in revising the manuscript which I now recommend for publication in Journal of fungi but also suggest minor revisions to manuscript (indicated on manuscript).
Please take into consideration to add a few concluding sentences to the end of manuscript.
L172-173: The authors state in author's respond to rev "we intentionally designed the experiments with soils that are enriched with the pathogen to ensure as high and uniform a disease pressure as possible".
For me, it is OK, I understand why you do the artificial inoculation with the pathogen but at the same time I still think you can't say: "on naturally infested field soils". Please change this sentence.
L254-256: “Consequently, relatively high standard error values resulted, and in most of those tests, no statistically significant difference could be measured in comparison to the control. This is also true for indoor growth experiments, whereas achieving a uniform infection with this particular pathogen Israeli strains, is challenging”.
My suggestion is that the authors should increase the number of replicates in order to reduce standard error as much as possible.

Author Response
Responses to Reviewer 2’s comments
We thank the reviewer for investing time and effort, which contributed to this manuscript. The helpful and necessary remarks and suggestions improved this scientific paper and made it more accurate, clear, and focused. Thank you.
The reviewer comments and remarks appeared on the manuscript document:
Line 24 - in vitro (corrected to Italic).
Line 56 –Verticillium dahlia - corrected to Verticillium dahliae.
Line 94 - cultivar Megaton cv. – corrected to cultivar Megaton.
Line 106 – indeed, this reference was a duplicate. Reference 29 was corrected to 8, and all reference list was updated accordingly.
Table 1 – all cultivars’ names were corrected as advised.
Lines 172-173 – we correct this, as will be detailed below.
Lines 255-256 – True, as answered in our comments below, this is a very good recommendation, and we will apply it in our future studies. The following explanation was added to the Discussion (Lines 436-438): “Also, the objective difficulty of achieving a uniform infection with this particular pathogen Israeli strains may be reduced if the number of plants used in each assay will be increased.”
Table 4 – all cultivars’ names were corrected as advised.
Line 289 – We correct “sweet” to “Sweet”, as advised.
All references were carefully checked to avoid typographic and editing mistakes.
The reviewer comments in the review letter:
Please take into consideration to add a few concluding sentences to the end of manuscript.
A conclusion section was added as suggested, at the end of the paragraph (lines 457-470).
L172-173: The authors state in author’s respond to rev “we intentionally designed the experiments with soils that are enriched with the pathogen to ensure as high and uniform a disease pressure as possible”.
For me, it is OK, I understand why you do the artificial inoculation with the pathogen but at the same time I still think you can’t say: “on naturally infested field soils”. Please change this sentence.
This is a correct remark. In the growth chamber experiment with soil samples taken from commercial fields, having a long history of M. maydis contamination: the Kibbutz Amir field soil and the Kibbutz Neot Mordechai, we didn’t add artificial inoculation. The paragraph was corrected, and the following sentence was added to clarify this point: “The direct examination of those soils impact on late wilt disease symptoms progression was done without the enrichment with the pathogen (done in the other growth chamber experiment).” (lines 175-177)
L254-256: “Consequently, relatively high standard error values resulted, and in most of those tests, no statistically significant difference could be measured in comparison to the control. This is also true for indoor growth experiments, whereas achieving a uniform infection with this particular pathogen Israeli strains, is challenging”.
My suggestion is that the authors should increase the number of replicates in order to reduce standard error as much as possible.
The reviewer is correct, this is a very good recommendation, and we will apply it in our future studies. The following explanation was added to the Discussion (Lines 436-438): “Also, the objective difficulty of achieving a uniform infection with this particular pathogen Israeli strains may be reduced if the number of plants used in each assay will be increased.”
This manuscript is a resubmission of an earlier submission. The following is a list of the peer review reports and author responses from that submission.
Round 1
Reviewer 1 Report
Dear Authors,
The fungus Magnaporthiopsis maydis can cause late wilt of Zea mays (maize). It is known to severely affect sensitive maize (Zea mays) plants at the maturity stage. The work described in this manuscript is very interesting and could have important consequences to prevent the spread of this disease and achieve effective control of it. This pathogen causes a rapid and sudden wilting, then an early diagnosis in plants is needed and may help to restrict disease spread. Due to the fact that infected seeds can carry the pathogen and spread the disease, molecular assays are important to recognize infected seeds and prevent spread to areas where the disease does not occur.
The organization of this work does not seem to me to be the most appropriate for the reader to understand it in its totality. Thus, the exposure of the results follows a different sequence compared to the description of the methodology used to obtain those results. Besides, there are some results that are simply not explained in materials and methods. There is too much methodology that addresses other references. This makes it hard to follow up on the work carried out. Both sections need to be carefully reworked for a better comprenhension
Regardless of the aforementioned, I think that the main problem with this work and that it made me take the decision to reject the publication of this manuscript in the Journal of Fungi is that I don't understand the advantage of the method described here to detect the pathogen. Your manuscript does present some interesting data and encourage you to use the comments to improve your manuscript.
- I consider that the authors does not provide a fast and efficient method, so it is required to await the development of symptoms (22 DAS) in the Megaton cultivar which is highly susceptible in a "growth chamber seedling test". Even more time is required to detect it in field trials (up to 69 DAS). The Megaton cv qPCR assays were only validated in the field for one year. I suggest validation with at least one more year of testing and quantifying the infection in the field.
Another essential issue to draw conclusions from the results is the statiticasl nanlysis. I have observed very high standard errors in the qPCR calculations, especially in the Megaton cv values. Unfortunately, this represents a serious problem in order to accept these results as reliable. I think you should increase the number of replications
- The sensitivity of the method is not specified. It is crucial to determine the minimum concentration of the pathogen in the soil in order to be able to detect the pathogen in the plant. The authors concluded that the advantage of this method is its ability to detect even low amounts of the fungus in soil. However, I think that the assays conducted here do not enable the authors to conclude this, as I cannot find any quantification of the pathogen that relates it to the results. There're not any assays of sensitivity.
- In "Materials and methods", the authors describe pot assays using soil naturally infected with the pathogen, however, the authors further describe that these soils are enriched with the pathogen to ensure high and uniform disease pressure as possible. This situation will never be encountered under natural conditions. So how does the method afford us the ability to differentiate natural infection from artificial infection?
- What is the method of DNA extraction? The DNA extraction method of fungus DNA from plants is essential to obtain satisfactory results in any test involving a PCR
- Have you checked the interaction that can occur with the plant's DNA? and Have you determined gDNA quantification and purity? gDNA integrity was checked
- You have the required controls in qPCR: with pathogen DNA only, with maize DNA and without any type of DNA
- Furthermore, a TaqMan probe assay has been developed to target M. maydis (Campos et al 2020) This newly developed TaqMan methodology was demonstrated in a field experiment through the screening of potentially infected maize roots, revealing a high specificity and proving to be a suitable tool to ascertain Fusarium spp. and M. maydis infection in maize. Its high sensitivity makes it very efficient for the early diagnosis of the diseases and also for certification purposes. However, the SYBR®Green qPCR assay used here for the detection of M. maydis DNA in maize Megaton cv however, it was not tested whether the method used was specific to M. maydis, or if it also could target Fusarium spp. It is known whether the megaton cultivar is susceptible to infection by Fusarium spp.
Best Regards
Reviewer 2 Report
The manuscript focusses one of the majors contains to maize production in several countries. The approach used is interesting and the methods are appropriate.
My major concern regards the experimental design of the experiments and the total number of samples analysed, that needs to be better clarified by the authors.
- Line 56: There are several other even more actual references for detection of plant fungi pathogens, including for M. maydis. Please include other references.
- line 147: include ‘were’. ‘These seeds were incubated…’
- line 148: 'designate' it is not the correct word here; 'were used' maybe!?
- line 159: I think you refer to DNA extraction instead of DNA purification.
- line 162: you refer to ‘Five plants of each pot were sampled, the samples were combined, and the total fresh weight was adjusted to 0.7 g and regarded to be one replicate.’ How many pots you consider for each cultivar? only one? Do you mean a bulked sample with the 5 plants from each pot for qPCR? How many biological replicates for each cultivar?
- How many samples were considered in the field experiments (point 3.4.) on qPCR analysis, for each cultivar and for each plot?
- What was the amount of DNA used on qPCR analysis?
- In the point ‘Molecular diagnose’ from Materials and methods, please refer to the experimental design and total number of samples analysed.
- The points in materials and methods are 2.1, 2.2… and not 3.1, 3.2….
- In the results you compared the maize plants results to the pathogen DNA variations within the roots of Setaria viridis. I cannot find any reference to this plant species in Material and Methods.
- I would like to know if the authors tested the qPCR methodology using directly soil DNA samples. I think that it would be important to compare soil and plant samples. The qPCR approach being a highly sensitive methodology will for sure amplify the M. maydis even from low DNA concentrated samples from the soil. I would like to have the authors opinion on that.
Responses to Reviewer 1’s comments
We thank the reviewer for investing substantial work that contributes significantly to this manuscript. His/her remarks and suggestions improved this scientific paper remarkably and made it more accurate, clear, focused and well-structured. Your contribution is greatly appreciated.
General comments:
The organization of this work does not seem to me to be the most appropriate for the reader to understand it in its totality. Thus, the exposure of the results follows a different sequence compared to the description of the methodology used to obtain those results.
We organized the results according to the following logic: • First, we presented the annual field assessments of new maize varieties for determination of their degree of resistance to late wilt disease. This assessment enables the discovery of a newly developed hybrid, Megaton cv. (Figures 1, 2). • Then, we used this hybrid to evaluate the degree of infection in two soil samples from commercial fields having a long history of M. maydis contamination. However, relying solely on symptoms evaluation can lead to inconsistent results (Figure 3). • Thus, a molecular quantitative real-time PCR (qPCR) method was introduced and used, first in a seven-day in vitro seed assay (Figures 4, 5), and then in growthchamber potted plants up to 30 days (Figures 6, 7, 8).
The Materials and Methods section structure was reworked to better reflect this data presentation logic: Section 3.4. (Field observation for assessing maize cultivars’ resistance/susceptibility to late wilt disease) was moved to the beginning of the Materials and Methods section. The sections in Materials and Methods are now ordered as follows:
2.1. Field observation for assessing maize cultivars’ resistance/susceptibility to late wilt disease 2.2. Fungal isolates and growth conditions 2.3. In vitro seed infection 2.4. Growth chamber seedlings assay 2.5. Molecular diagnosis 2.6. Statistical analyses
Besides, there are some results that are simply not explained in materials and methods.
We carefully checked and verified that all the data presented in the Results section were explained in the Materials and Methods section. Indeed, in one case, some details were missing. The following explanation was added to the text (lines 171-179):
“Maize sprout susceptibility assays were performed under controlled conditions and aimed at achieving two goals. The first goal was to identify variations in the severity of the symptoms of plants cultivated on naturally infested field soils taken from two different locations. Soil samples were taken from commercial fields having a long history of M. maydis contamination: the Kibbutz Amir field soil [8,30] and the Kibbutz Neot Mordechai field soil [15,27]. Also, an evaluation of the symptoms was made to study differences among selected maize cultivars and between those cultivars and S. viridis (green foxtail), a recently discovered new host of this fungus [18]. This is important to identify the most appropriate test plant for the soil assay. Our second aim was to determine the optimal cultivar and growth period duration that will be used to inspect susceptibility to late wilt infested soils.”
There is too much methodology that addresses other references. This makes it hard to follow up on the work carried out.
We double-checked this and ensure that all necessary information is now included in the text:
• Lines 145-147, the sentence was rephrased: “To examine the pathogens’ ability to infect maize seeds in vitro, seeds were inoculated with the fungus and then tested for the fungus’ DNA presence in the inner tissues in a previously developed method [8].” • The in vitro seed infection technique, used in this study, is now detailed thoroughly (lines 217-229).
• The molecular DNA isolation and extraction details were added to the text (lines 217-229): “After grinding the tissue samples with 4 mL CTAB buffer (0.7 M NaCl, 1% cetyltriammonium bromide, 50 mM Tris-HC1 pH 8.8, 10 mM EDTA and 1% 2-mercaptoethanol), 1.2 mL from this mixture was kept for 20 min at 65°C. For DNA extraction, we used the Eppendorf Centrifuge 5810 R. The samples were centrifuged for 5 min at 25°C at 14,000 rpm. The upper lysate part (usually 700 µL) was extracted with an equal volume of chloroform/isoamyl alcohol (24:1). Following mixing by vortex, the mixture was centrifuged again at 14,000 rpm for 5 min at 25°C. The chloroform/isoamyl-alcohol extraction was repeated twice. The supernatant (approximately 300 µl) was then moved to a new Eppendorf tube and mixed with cold isopropanol (2:3). The DNA suspension was mixed gently by inverting the tube several times, maintained for 20-60 min at 20°C, and centrifuged (14,000 rpm for 20 min in 4°C). The precipitated DNA isolated was resuspended in 0.5 mL 70% ethanol. After additional centrifugation (14,000 rpm at 4°C for 10 min), the DNA pellet was dried in a sterile hood overnight. The DNA was suspended by the addition of 100 µL of HPLC-grade water and maintained at 20°C until use in the qPCR reactions.”
Both sections need to be carefully reworked for better comprehension.
We made our best effort to address this concern. We hope that this major revision of the manuscript, which was made with our full attention to both reviewers’ suggestions and remarks, improves the manuscript’s scientific accurateness and clarity.
The main problem with this work is that I don’t understand the advantage of the method described here to detect the pathogen. Your manuscript does present some interesting data and encourage you to use the comments to improve your manuscript.
The advantage of using the methodology described in this work to detect M. maydis was detailed throughout the text in several places:
• Abstract, lines 14-15: “However, no soil assay is available today that can detect M. maydis infestation and study its prevalence.” • Abstract, Lines 25-26: “These results now enable the use of this highly susceptible testing plant to validate the presence of the maize late wilt pathogen in infested soils and to evaluate the degree of its prevalence.” • Introduction, lines 59-66: “However, in the case of M. maydis, the goal of developing an effective soil assay has yet to be achieved. This assay may be challenging to develop due to low quantities of the fungus in the soil and the scattered nature of the disease (that spreads in patches in the field). To make this task even harder, DNA extracted from soil samples may include PCR inhibitors [20]. With the gradual canceling of soil disinfecting using methyl bromide [21], lengthier intervals are needed between soil treatments, using alternative fungicides, and sowing. This challenge, together with the increase in production of organic cultivars, necessitates an urgent need for an accurate and rapid method for the determination of M. maydis soil inoculum level.” • Introduction, lines 69-76: “Also, as noted earlier by Saleh et al. (2003), M. maydis recovery from plant material is challenging. This difficulty is true even with heavily inoculated plant tissues, due to the pathogen’s slow development and the relatively high prevalence of other, more fast-growing fungi, specifically Fusarium spp. Indeed, late wilt disease is often accompanied by secondary plant pathogenic fungi contamination, enhancing the stem symptoms. Such pathogenic fungi opportunists are Fusarium verticillioides causing stalk rot [15] and Macrophomina phaseolina causing charcoal rot [17]. We invested considerable efforts in M. maydis isolation from infested commercial fields with a long history of late wilt harsh damages, but with minor success [25].” • Introduction, lines 93-96: “The present work focuses on the application of the RealTime PCR (qPCR) protocol for evaluating the potential use of the maize cultivar Megaton cv. (Hazera Seeds Ltd., Berurim MP Shikmim, Israel) compared to other cultivars for the detection and estimation of soils infested with M. maydis.” • Results, lines 295-303: “A dedicated set of experiments aimed at adjusting the seedling assay under controlled conditions will allow us to make a relatively rapid determination of the degree of soil infestation with the late wilt causal agent. To this end, we used the newly discovered Megaton cv. as a potential check genotype and measured its disease severity in response to the soils’ prevalence of the pathogen (Fig. 3).” • Discussion, lines 395-410: “There is an urgent need to develop a more rapid, accurate and specific assay to determine the degree of M. maydis infestation in suspicious soils. Today, no such method exists, and relying on traditional media plant protocols to isolate the pathogen from the ground has several disadvantages that make this solution impractical. A limitation of this technique includes the difficulty in isolating the pathogen from the soil due to its scattered nature and low prevalence. Also, the pathogen has a relatively slow growth rate of about 1 cm per day on PDA media plates and optimal temperature [15], and it is considered a poor competitor in a mixture of microorganisms. Thus, from the soil mycoflora existing in the sample, rapid-growth fungi such as Fusarium spp. will most probably take over and cover the plate [34]. This limitation is true even if antibiotics are added to the medium [25]. Hence, the media plate technique to isolate the pathogen directly from the soil is time-consuming and produces inconsistent results. The seedling pathogenicity inspection is a preferable method for soil testing. It provides more reliable and relatively rapid results while minimizing the risk of missing M. maydis identification (receiving false-negative results). To this end, we developed and validated a qPCR-based soil bioassay that facilitates detecting and tracking the pathogen in a testing maize plant, the Megaton cv.” • Discussion, lines 423-443: “The advantage of the proposed bioassay using a susceptible cultivar and a qPCR-based method is its ability to detect the fungus even if it scattered in the soil. This assay enables us to isolate and enrich the maize late wilt pathogen from the soil using a trap plant. The use of this bioassay is essential for study M. maydis distribution and to provide an estimation of its infestation degree in commercial fields. Such data are necessary for decisionmaking among growers to reduce disease damage and provide risk assessment and insurance companies with vital information that will help them to support the farmers. Identifying the fungus will allow the farmer to make decisions about sowing time, choosing the appropriate maize cultivar for planting, the application of disease management strategies, the implementation of a prevention plan, and more. All these means may also help quarantine the infected areas and prevent the disease from spreading to new fields. The proposed assay may also be used to evaluate the virulence of different M. maydis isolates. Indeed, different M. maydis strains differ in morphology and mode of infection [35]. For example, four different lines of M. maydis isolated in Egypt showed different abilities to establish and infect corn plants [31,34,36]. In southern Portugal and Spain, an analysis of 14 M. maydis isolates was performed using 32 different maize varieties [37]. One of the isolates was extremely virulent and caused intense symptoms that included significant decreases in the fresh weight of both above-ground parts and roots.”
Specific comments:
I consider that the authors does not provide a fast and efficient method, so it is required to await the development of symptoms (22 DAS) in the Megaton cultivar which is highly susceptible in a “growth chamber seedling test”.
The reviewer is correct, the suggested bioassay is only relatively faster and efficient compared to traditional methods. We rephrased the following sentences to better and more accurately express this:
• “A dedicated set of experiments aimed at adjusting the seedling assay under controlled conditions will allow us to make a relatively rapid determination of the degree of soil infestation with the late wilt causal agent.” (lines 295-297) • “There is an urgent need to develop a more rapid, accurate and specific assay to determine the degree of M. maydis infestation in suspicious soils. (lines 395-396) • “The seedling pathogenicity inspection is a preferable method for soil testing. It provides more reliable and relatively rapid results while minimizing the risk of missing M. maydis identification (receiving false-negative results).” (lines 406408)
Even more time is required to detect it in field trials (up to 69 DAS).
The field trial described here is not part of the M. maydis soil bioassay. It was only used to examine new maize cultivars’ resistance/sensitivity to late wilt disease (see lines 103107). During this annual routine, the new Megaton cv. was discovered. This hypersusceptible cultivar may now be used in the soil bioassay as a test plant.
As mentioned in the Discussion (lines 412-415): “The finding that Megaton is so susceptible in the field encouraging to screen diverse inbreeds by the field screening method, which may lead to an identified more ideal trap plants that would allow pathogen detection reproducibly within days of planting.”
The Megaton cv qPCR assays were only validated in the field for one year. I suggest validation with at least one more year of testing and quantifying the infection in the field.
As explained above, the field assay is not part of the M. maydis soil bioassay. The Megaton cv. qPCR assays were validated in a series of trials that includes in vitro seeds and in vivo sprouts pathogenicity assays. As mentioned in Materials and Methods, the whole experiment was repeated twice for both assays, the in vitro seed infection and the growth chamber seedlings assay, and similar results were obtained.
Another essential issue to draw conclusions from the results is the statistical analysis. I have observed very high standard errors in the qPCR calculations, especially in the Megaton cv values. Unfortunately, this represents a serious problem in order to accept these results as reliable. I think you should increase the number of replications.
This is indeed an important aspect that should be addressed. Field pathogenicity trials that rely on natural soil infestation have highly variable results. Even in heavily infested fields, the spreading of the pathogen is not uniform. The pathogen is scattered in small quantities in the soil, and the disease spreading in the field is not uniform (see, for example, Degani et al., Agronomy 2019, 9, 181, Figure 1). Moreover, the variables are subject to changes in environmental conditions.
The qPCR method we used is very sensitive and capable of detecting variations in the amount of the pathogens’ DNA inside the host plant tissues with a million-fold difference (see, for example, Degani et al., PloS One 2018, 13, e0208353). Nevertheless, in fields that have moderate or minor disease outbursts, some of the measurements resulted in zero values. This is also true for indoor growth experiments, whereas achieving a uniform infection with this particular pathogen in Israeli strains is challenging (see Degani et al., Plant Disease, 2019, 103, 238-248).
This explanation is presented in Materials and Methods (lines 251-256): “In the field trials’ qPCR-based-molecular DNA tracking there is on the objective difficulty to achieve uniform repeats, due to changes in environmental conditions, and the non-uniform spreading nature of the late wilt disease pathogen [33]. Consequently, relatively high standard error values resulted, and in most of those tests, no statistically significant difference could be measured in comparison to the control. This is also true for indoor growth experiments, whereas achieving a uniform infection with this particular pathogen Israeli strains, is challenging [8].”
The sensitivity of the method is not specified. It is crucial to determine the minimum concentration of the pathogen in the soil in order to be able to detect the pathogen in the plant. The authors concluded that the advantage of this method is its ability to detect even low amounts of the fungus in soil. However, I think that the assays conducted here do not enable the authors to conclude this, as I cannot find any quantification of the pathogen that relates it to the results. There’re not any assays of sensitivity.
The reviewer is correct. We only argue that the qPCR detection is sensitive, however, it should be clarified that the sensitivity of the bioassay was not determined in this work and should be the focus of a subsequent study.
Therefore, we rewrote this sentence to avoid this interpretation: “The advantage of the proposed bioassay using a susceptible cultivar and a qPCR-based method is its ability to detect the fungus even if it scattered in the soil.” (lines 423-424)
Also, the following explanation was added to the Discussion: “Together with these advantages, it should be noted that the sensitivity of the bioassay method presented in this work was not specified. It would be beneficial to determine the minimum concentration of the pathogen in the soil that would allow the detection of the pathogen in the plant. This should be the focus of a subsequent study.” (lines 433-436)
In “Materials and methods”, the authors describe pot assays using soil naturally infected with the pathogen, however, the authors further describe that these soils are enriched with the pathogen to ensure high and uniform disease pressure as possible. This situation will never be encountered under natural conditions. So how does the method afford us the ability to differentiate natural infection from artificial infection?
Indeed, the development and calibration of the growth chamber seedlings assay required two steps:
1. Evaluate the Megaton cv. late wilt sensitivity in comparison to two other wellstudied representative sensitive, sweet maize cultivars. 2. Inspect the Megaton cv. susceptibility in potted sprouts at different ages up to the age of 40 days.
During this calibration, we intentionally designed the experiments with soils that are enriched with the pathogen to ensure as high and uniform a disease pressure as possible.
Of course, the new bioassay will be used to identify late wilt infested soils without any additional inoculation.
What is the method of DNA extraction? The DNA extraction method of fungus DNA from plants is essential to obtain satisfactory results in any test involving a PCR.
As specified above, the molecular DNA isolation and extraction details were added to the text (lines 214-229).
Have you checked the interaction that can occur with the plant’s DNA? and Have you determined gDNA quantification and purity? gDNA integrity was checked. You have the required controls in qPCR: with pathogen DNA only, with maize DNA and without any type of DNA.
The reviewer is correct; this is indeed an important issue that should be clarified.
The specific M. maydis qPCR detection was just recently validated, approved and published (Degani et al., Plant Disease, 2019, 103, 238-248). We have used this qPCR method repeatedly in several additional works (see Degani et al., PloS One 2018, 13, e0208353; Degani et al., Agronomy 2019, 9, 181; Dor and Degani, Plants 2019, 8; Degani et al., Microorganisms 2020, 8, Degani et al., Journal of Fungi 2020, 6). The same protocol with some adjustments is used worldwide in the scientific community for a similar purpose (identifying pathogens’ DNA inside host tissues).
We routinely run positive control in the qPCR reactions that include DNA from a PDA growth culture of M. maydis with both the cox1 and M. maydis (A200a) primers and get measurable and relatively constant results.
We calculated the relative DNA levels using the mean ΔCt value (threshold cycle). The method we used for calculating relative DNA abundance from the quantification cycle (Cq) values obtained by the qPCR analysis is explained well in the following citation: Haimes, J., and M. Kelley. “Demonstration of a ΔΔCq calculation method to compute relative gene expression from qPCR data” Thermo Scientific Tech Note 1 (2010).
The following explanation is given in the text: “The target A200a, M. maydis-specific DNA, was evaluated against a reference “housekeeping” gene – the mitochondriacytochrome c oxidase, COXI gene (sequences in [33]). This reference gene encoding the eukaryotic mitochondria respiratory electron transport chain’s last enzyme was used to normalize the amount of DNA. Calculating the relative DNA abundance was according to the ΔCt model [41]. The same efficacy was assumed for all samples. All amplifications were performed in triplicate.” (lines 240-245).
Furthermore, a TaqMan probe assay has been developed to target M. maydis (Campos et al 2020) This newly developed TaqMan methodology was demonstrated in a field experiment through the screening of potentially infected maize roots, revealing a high specificity and proving to be a suitable tool to ascertain Fusarium spp. and M. maydis infection in maize. Its high sensitivity makes it very efficient for the early diagnosis of the diseases and also for certification purposes. However, the SYBR®Green qPCR assay used here for the detection of M. maydis DNA in maize Megaton cv however, it was not tested whether the method used was specific to M. maydis, or if it also could target Fusarium spp. It is known whether the megaton cultivar is susceptible to infection by Fusarium spp.
The SYBR®Green qPCR assay used here for the detection of M. maydis DNA in maize Megaton cv. is based on the A200 resultant amplified sequence. This sequence is a major part of a larger AFLP fragment that had been proven earlier to be species-specific by Saleh et al. (2003)1 and Zeller et al. (2000)2. The specific M. maydis detection was reported by us for PCR in 2013 (Drori et al., Phytopathologia Mediterranea, 2013), and was just recently validated for qPCR, approved and published (Degani et al., Plant Disease, 2019). Moreover, we tested and used the specific molecular method in seven additional publications3 in leading scientific journals.
These species-specific primers were tested and validated in the past decade in our lab in in vitro experiments with a single pathogen inoculation (pure cultures, seed assay, detached root assay) in seedlings in a growth chamber and in full growth season experiments in a greenhouse. They were also used in commercial fields with naturally infested soils, as presented in this work. All these experiments were conducted under strict experimental design conditions and were well controlled. Negative uninfected controls repeatedly resulted in axenic tissues and zero levels of detection with the molecular method. Moreover, the levels of M. maydis DNA measured throughout the season were in correlation with the maize late wilt disease-specific symptoms severity (including dehydration symptoms and vascular tissue occlusion) and outcome (their effect on plant phenological development, yield production and yield quality).
1 Saleh, A. A. et al. Amplified fragment length polymorphism diversity in Cephalosporium maydis from Egypt. Phytopathology 93, 853-859 (2003). 2 Zeller, K. A., Jurgenson, J. E., El-Assiuty, E. M. & Leslie, J. F. Isozyme and amplified fragment length polymorphisms from Cephalosporium maydis in Egypt Phytoparasitica 28, 121-130 (2000). 3 Degani, O. & Cernica, G., Advances in Microbiology (2014), Degani et al., Phytoparasitica (2014), Degani et al., Physiology and Molecular Biology of Plants (2015), Degani et al. PloS One (2018), Degani et al., Agronomy (2019), Dor, S. & Degani, Plants (2019), Degani et al., Microorganisms (2020).
In addition, we tested the ability of the A200 primers to detect Fusarium sp. (probably F. verticillioides) isolated from commercial field disease maize plants4. The molecular PCR assay was clearly able to identify M. maydis, but not the Fusarium isolate.
This being said, the TaqMan probe assay is indeed a highly specific and sensitive diagnostic tool, which makes it very efficient for the early diagnosis of late wilt of maize causal agent. Unfortunately, when we developed the soil bioassay over the last two years, this recent technique was not available in our lab. It would be very interesting to study a combination of the Megaton cv. pot bioassay together with the TaqMan probe assay in order to achieve an improved soil assay with maximum efficiency.
The following paragraph was added to the end of the Discussion (lines 450-456): “A TaqMan probe assay was developed recently to target M. maydis [21]. This newly developed methodology was demonstrated in a field experiment through the screening of potentially infected maize roots, revealing a high specificity and proving to be a suitable tool to ascertain M. maydis infection in maize. Its high sensitivity makes it very efficient for the early diagnosis of the diseases and also for certification purposes. It will be most interesting to study the combination of the Megaton cv. pot bioassay together with the TaqMan probe assay in order to achieve an improved soil assay with maximum efficiency and sensitivity.”
Responses to Reviewer 2’s comments
We would like to express our sincere appreciation to the reviewer for the important and helpful suggestions and advice. The time and effort invested are greatly appreciated, and without a doubt, contributed to the manuscript and significantly improved it. Thank you.
General comments:
My major concern regards the experimental design of the experiments and the total number of samples analysed, that needs to be better clarified by the authors.
This is indeed an important aspect that should be addressed. As explained in Materials and Methods, all experiments were conducted with six independent repeats. Each of the experiments was repeated twice, with similar results obtained in both repeats. Clear differences were measured between the results with standard errors, which allow measuring significant differences (p<0.05, ANOVA) between the results and the control (except for the qPCR results).
Regarding the variability of the qPCR results, field pathogenicity trials that rely on natural soil infestation have highly variable results. Even in heavily infested fields, the spreading of the pathogen is not uniform. The pathogen is scattered in small quantities in the soil and the disease spreading in the field is not uniform (see, for example, Degani et al., Agronomy 2019, 9, 181, Figure 1). Moreover, the variables are subject to changes in environmental conditions.
4 Drori, R. Involvement of Harpophora maydis in late wilt disease of sweet corn: characterization of the disease cycle and identifying means of control. Master’s in Science of Agriculture thesis, Hebrew University of Jerusalem (2010).
The qPCR method we used is very sensitive and capable of detecting variations in the amount of the pathogens’ DNA inside the host plant tissues with a million-fold difference (see, for example, Degani et al., PloS One 2018, 13, e0208353). Nevertheless, in fields that have moderate or minor disease outbursts, some of the measurements resulted in zero values. This is also true for indoor growth experiments, whereas achieving a uniform infection with this particular pathogen in Israeli strains is challenging (see Degani et al., Plant Disease, 2019, 103, 238-248).
This explanation is presented in Materials and Methods (lines 251-256): “In the field trials’ qPCR-based-molecular DNA tracking there is on the objective difficulty to achieve uniform repeats, due to changes in environmental conditions, and the non-uniform spreading nature of the late wilt disease pathogen [33]. Consequently, relatively high standard error values resulted, and in most of those tests, no statistically significant difference could be measured in comparison to the control. This is also true for indoor growth experiments, whereas achieving a uniform infection with this particular pathogen Israeli strains, is challenging [8].”
Details of corrections:
Line 56: There are several other even more actual references for detection of plant fungi pathogens, including for M. maydis. Please include other references.
We agree, the paragraph was rewritten and now includes new and more actual references (lines 53-58): “Another approach to providing a faster technique of fungal pathogens’ detection and quantification in the soil is the use of polymerase chain reaction (PCR) molecular identification [19]. Customary PCR tests using nested or single amplifications have been suggested for detecting other fungi (summarized, for example, for Verticillium dahlia by [20]). A TaqMan probe assay has been developed lately to target M. maydis [21] and could be used for the same purpose, as demonstrated for the potato pathogen, Synchytrium endobioticum [22].”
line 147: include ‘were’. ‘These seeds were incubated…’
Corrected as advised.
line 148: ‘designate’ it is not the correct word here; ‘were used’ maybe!?
Corrected as advised.
line 159: I think you refer to DNA extraction instead of DNA purification.
Right, corrected as advised.
line 162: you refer to ‘Five plants of each pot were sampled, the samples were combined, and the total fresh weight was adjusted to 0.7 g and regarded to be one replicate.’ How many pots you consider for each cultivar? only one? Do you mean a bulked sample with the 5 plants from each pot for qPCR? How many biological replicates for each cultivar?
Each experimental group (cultivar) comprised six biological replicates (pots). Each pot was sown with five seeds. Indeed, a bulked sample with the five plants from each pot was used for the qPCR.
The following paragraph was rephrased to better and more accurately express this (lines 180-182): “Each experimental group comprised six biological replicates (pots). Each pot was sown with five seeds. Maize hybrids Megaton cv. and Royalty cv., and S. viridis were inoculated with M. maydis as previously described [25].”
How many samples were considered in the field experiments (point 3.4.) on qPCR analysis, for each cultivar, and for each plot?
Each cultivar was seeded in a different row and the rows arranged in the field using a randomized complete block design. Each row was 12 m long and included 8 maize plants m-1 (approximately 96 plants per row) (see lines 117-119).The dehydration assessment was conducted for all the plants of each cultivar by calculating the percentage of plants having typical maize late dehydration symptoms (see lines 127130).
Regarding the qPCR analysis, the following explanation was added as per your recommendation: “For each cultivar, the qPCR analysis was conducted on three representative replications (plants).“ (lines 134-135)
What was the amount of DNA used on qPCR analysis? In the point ‘Molecular diagnose’ from Materials and methods, please refer to the experimental design and total number of samples analysed.
We calculated the relative DNA levels using the mean ΔCt value (threshold cycle). The method we used for calculating relative DNA abundance from the quantification cycle (Cq) values obtained by the qPCR analysis is explained well in the following citation: Haimes, J., and M. Kelley. “Demonstration of a ΔΔCq calculation method to compute relative gene expression from qPCR data” Thermo Scientific Tech Note 1 (2010).
The following explanation is given in the text: “The target A200a, M. maydis-specific DNA, was evaluated against a reference “housekeeping” gene – the mitochondriacytochrome c oxidase, COXI gene (sequences in [33]). This reference gene encoding the eukaryotic mitochondria respiratory electron transport chain’s last enzyme was used to normalize the amount of DNA. Calculating the relative DNA abundance was according to the ΔCt model [41]. The same efficacy was assumed for all samples. All amplifications were performed in triplicate.” (lines 240-245).
The points in materials and methods are 2.1, 2.2… and not 3.1, 3.2….
Right, corrected as advised.
In the results you compared the maize plants results to the pathogen DNA variations within the roots of Setaria viridis. I cannot find any reference to this plant species in Material and Methods.
Thank you for this comment; the information about Setaria viridis was added to Section 2.4. (Growth chamber seedlings assay):
“Also, an evaluation of the symptoms was made to study differences among selected maize cultivars and between those cultivars and S. viridis (green foxtail), a recently discovered new host of this fungus [18]. This is important to identify the most appropriate test plant for the soil assay. Our second aim was to determine the optimal cultivar and growth period duration that will be used to inspect susceptibility to late wilt infested soils. Each experimental group comprised six biological replicates (pots). Each pot was sown with five seeds. Maize hybrids Megaton cv. and Royalty cv., and S. viridis were inoculated with M. maydis as previously described [25].” (lines 175-182)
I would like to know if the authors tested the qPCR methodology using directly soil DNA samples. I think that it would be important to compare soil and plant samples. The qPCR approach being a highly sensitive methodology will for sure amplify the M. maydis even from low DNA concentrated samples from the soil. I would like to have the authors opinion on that.
Indeed, if we could use the qPCR technique to measure the soil infestation degree directly, this would be very advantageous. However, despite the qPCR technique’s high sensitivity, direct examination of soil samples can be inconsistent. As previously mentioned in the Introduction, lines 59-62: ”However, in the case of M. maydis, the goal of developing an effective soil assay has yet to be achieved. This assay may be challenging to develop due to low quantities of the fungus in the soil and the scattered nature of the disease (that spreads in patches in the field). To make this task even harder, DNA extracted from soil samples may include PCR inhibitors [23].”
Also, it should be remembered that the qPCR technique is based on very small samples (0.7 g), and even if those samples are extracted from a larger soil bulk, they still poorly represent the selected field. In contrast, pots contain 2 kg of soil (a nearly 3,000 times larger sample quantity), which made this assay a better representative of the field.